# Innovative Sensor-Based Approaches for Assessing Neurodegenerative Diseases: A Brief State-of-the-Art Review

**DOI:** 10.3390/s25175476

**Published:** 2025-09-03

**Authors:** Ngozi D. Mbue, Fatemeh Tabei, Karen Williams, Kazeem Olanrewaju

**Affiliations:** 1Nelda C. Stark College of Nursing, Texas Woman’s University, Houston, TX 77030, USA; 2Department of Electrical Engineering/Computer Science, West Texas A & M University, Canyon, TX 79016, USA; ftabei@wtamu.edu; 3Department of Allied Health-Health Information Technology Program, Del Mar College, Corpus Christie, TX 78404, USA; kwilliams@delmar.edu; 4Department of Chemical Engineering, Prairie View A & M University, Prairie View, TX 77446, USA; kaolanrewaju@pvamu.edu

**Keywords:** neurodegenerative diseases, sensors, technologies, innovation, clinical practice

## Abstract

Sensor-based approaches are transforming the diagnosis and treatment of neurodegenerative diseases by offering more sensitive, non-invasive tools and are capable of real-time monitoring. Integrating advanced materials, nanotechnology, and artificial intelligence presents promise for earlier detection, enhanced disease management, and improved patient outcomes. From a clinical perspective, these technologies facilitate the shift toward precision medicine by enabling early intervention strategies, real-time treatment monitoring, and more refined patient stratification in practice and research contexts. This review provides an overview of recent advancements in sensor-based technologies aimed at enhancing the diagnosis and monitoring of neurodegenerative diseases (NDDs) such as Alzheimer’s and Parkinson’s, among others. Sensor-based technologies are adjunct tools and integral components of a next-generation framework for diagnosing, monitoring, and understanding neurodegenerative disorders.

## 1. Introduction

Neurodegenerative diseases (NDDs) are a group of disorders characterized by the progressive degeneration and death of nerve cells (neurons) in the brain and spinal cord [1]. These degenerations often result in a decline in brain function and neurological symptoms. Many NDD conditions eventually lead to dementia, which is projected to affect approximately 150 million people worldwide by 2050, imposing an economic burden of USD 10 trillion [2]. The exact causes of most neurodegenerative diseases remain unclear; however, factors such as genetic mutations, environmental toxins, viral infections, and aging contribute to the development of these conditions [2,3]. Symptoms of neurodegenerative diseases vary depending on the specific condition and its stage of progression. Common symptoms include the following: (1) cognitive decline (memory loss and confusion), (2) movement disorders (tremors, stiffness, and difficulty with coordination), (3) sensory problems (numbness, tingling), (4) muscle weakness, and (5) emotional changes (depression and anxiety) [4].

Common neurodegenerative diseases include the following: (1) Alzheimer’s disease (AD), one of the most prevalent neurodegenerative disorders, which leads to memory loss, confusion, and cognitive decline due to the buildup of amyloid plaques and tau tangles in the brain; (2) Parkinson’s disease (PD), which impacts movement and results in tremors, rigidity, and bradykinesia (slow movements) due to the loss of dopamine in the substantia nigra; (3) Huntington’s disease (HD), a genetic condition that results in uncontrolled movements, cognitive decline, and psychiatric symptoms; (4) Amyotrophic Lateral Sclerosis (ALS), also known as Lou Gehrig’s disease, which causes muscle weakness and paralysis from motor neuron degeneration; (5) Multiple Sclerosis (MS), an autoimmune condition that harms the myelin sheath surrounding nerve fibers, leading to neurological issues; and (6) Frontotemporal Dementia (FTD), which affects personality, behavior, and language due to degeneration in the frontal and temporal lobes [4,5,6,7,8].

Neurodegenerative diseases are progressive and typically worsen over time. There is currently no cure for most neurodegenerative diseases [2]. Treatment focuses on managing symptoms, slowing disease progression, and improving quality of life. Some treatments include medications to improve movement, cognition, and mood. Other treatment modalities include physical and occupational therapy, assistive devices, and lifestyle modifications such as exercise and a healthy diet. Research on new diagnoses and treatments for neurological diseases is ongoing. The current diagnosis of neurodegenerative disease typically involves a medical history and physical examination, neurological tests, imaging studies such as Magnetic Resonance Imaging (MRI) and Computed Tomography (CT) scans, and genetic testing [9,10]. See Figure 1: Systems biology approaches to understanding the host–microbiome interactions in neurodegenerative diseases [1].

Recent advancements in sensor-based technologies have significantly improved the assessment and diagnosis of neurodegenerative diseases [9]. These innovative approaches provide non-invasive, cost-effective, and accurate alternatives to traditional diagnostic methods. Some of these sensors include electrochemical biosensors, wearable sensors, gait analysis, artificial intelligence, digital biomarkers, and other innovative diagnostic devices such as handheld instruments capable of detecting ultra-low concentrations of disease markers from a single drop of blood [9,10]. For example, a palm-sized sensor developed by engineers at Monash University can quickly and painlessly diagnose Alzheimer’s and Parkinson’s diseases by detecting beta amyloids and tau proteins, offering instant results and enhancing accessibility to early diagnosis [11].

This review covers advances in innovative sensor technologies for monitoring NNDs, including wearable and sensor-based systems, as well as AI and machine learning applications. It also examines how remote monitoring and telemedicine are integrated into the diagnosis and management of these conditions. We discuss challenges, limitations, future directions, and research opportunities. The development of these advanced sensors underscores the crucial role of sensor-based technologies in enhancing the assessment and management of neurodegenerative diseases, thereby enabling more personalized and timely interventions. See Table 1: Key systems biology approaches in host–microbiome–NDD interaction [12].

**Figure 1 sensors-25-05476-f001:**
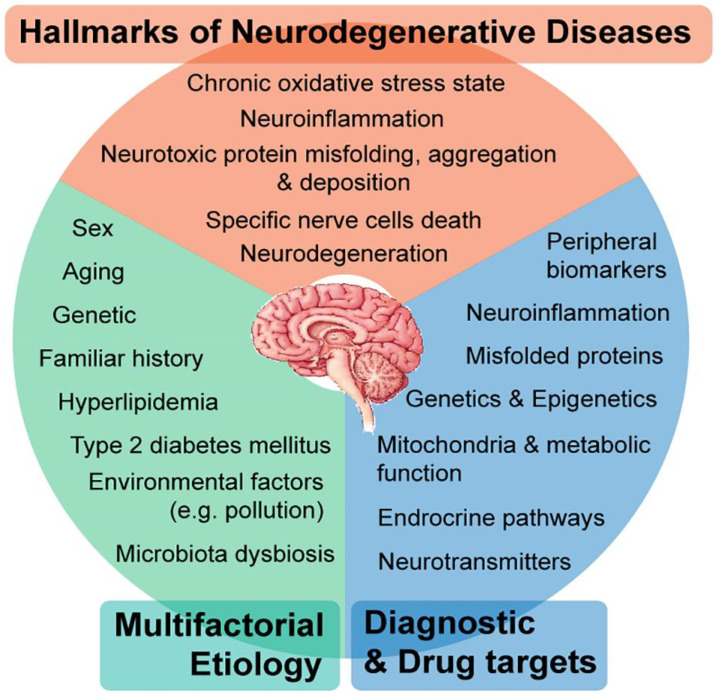
Systems biology approaches to understanding the host–microbiome interactions in neurodegenerative diseases/frontiers [1,12].

**Table 1 sensors-25-05476-t001:** Key systems biology approaches in host–microbiome–NDD interaction [12].

Approach/Component	Description/Role
Holistic Systems Biology	Integrates multi-layered biological data (e.g., genomics, metabolomics) to model the interplay between host and microbiome in neurodegeneration.
Microbiota–Gut–Brain Axis Pathways	Examines direct and indirect mechanisms (metabolites, immune modulators, barrier effects) through which the microbiome influences the CNS.
Microbial Dysbiosis Evidence	Surveys shift in gut microbial community—particularly in Alzheimer’s and Parkinson’s disease—and explores how these imbalances contribute to NDDs.
Dietary Modulation Strategies	Explores diet-based interventions to stimulate neuroprotective microbial metabolite production by modifying microbial composition.
Genome-Scale Metabolic Models (GEMs)	Uses GEMs to simulate microbe–microbe and host–microbe metabolic interactions, determining the microbiome’s impact on disease development or prevention.
Multi-Omics + GEMs for Personalized Diets	Proposes an integrative systems framework combining GEMs and omics data to design personalized, anti-inflammatory diets targeting gut microbiota in NDD prevention.

## 2. Advances in Technology for NND Monitoring

### 2.1. Emerging Technological Innovations

Innovative technologies are transforming healthcare through the use of artificial intelligence (AI), machine learning (ML), biotechnology, Internet of Things (IoT), blockchain, robotics, renewable energy, AR/VR, quantum computing, and 3D printing [13,14,15].

AI and ML: Used in natural language processing, imaging, speech recognition, predictive analytics, and personalization in healthcare and commerce. Trends include automation, ethical AI, deep learning, and explainable AI [16,17,18,19,20]. Biotechnology and Health Tech: Encompasses gene editing (CRISPR), personalized medicine, diagnostics, and telemedicine. Growth is seen in wearable health technology and advanced therapies [21,22,23]. Key trends include Internet of things (IoT) and advancements in collaborative robots (cobots) that operate alongside humans, as well as the increasing prevalence of remotely operated robots in hazardous environments [24,25,26].

IoT: Offers interconnected devices for healthcare monitoring, smart homes, and predictive analytics. Expanding 5G and IoT security are current priorities [27]. Blockchain: Supports secure, decentralized transactions, used in finance, supply chains, smart contracts, and identity verification. Trends include decentralized finance (DeFi) [28,29,30,31]. Robotics and Automation: Boosts productivity in manufacturing, healthcare, and hazardous environments, with collaborative robots (cobots) gaining popularity [32,33,34,35].

Renewable Energy and Sustainability: Solar, wind, biofuels, and EV adoption reduce reliance on fossil fuels [36,37,38]. Advanced Materials: Nanomaterials, smart materials, and biodegradable substances lead to new applications [39,40,41]. AR/VR: Used in gaming, education, and training, with growth in mixed reality and AR marketing [42,43].

Quantum Computing: Applies quantum mechanics to cryptography, optimization, and problem-solving [44,45]. 3D Printing: Enables rapid prototyping, custom manufacturing, and medical device innovation [46]. The convergence of AI, ML, biotechnology, and digital health is shaping precision medicine. Wearable and sensor-based devices, powered by AI, generate real-time physiological and behavioral data, improving early detection, progression tracking, and treatment personalization [47,48,49,50,51,52].

### 2.2. Wearable and Sensor-Based Monitoring Technologies

Wearables (e.g., fitness trackers, smartwatches, medical-grade devices) and embedded sensors (e.g., for heart rate, respiration, glucose, environmental monitoring) provide real-time health insights [53,54,55,56,57,58]. Integration with apps and cloud platforms facilitates remote monitoring, chronic disease management, and reduces the need for hospital visits [59]. Key trends include AI-driven predictive analytics for personalized recommendations [60,61,62].

Expanding the measurement of biochemical and physiological markers, along with improving comfort and design, enhances user adherence. However, challenges include data privacy and security (HIPAA compliance). Other issues include interoperability across devices and platforms, user adherence and engagement, as well as the accuracy and reliability of readings [63,64]. Wearables, coupled with AI/ML, create opportunities for early detection and intervention in neurodegenerative diseases, with predictive models identifying subtle physiological or behavioral shifts [65].

### 2.3. Emerging Sensor Technologies

Smartphone-integrated sensors and wearable inertial sensors achieve greater than 90% accuracy in gait detection and early Parkinson’s screening [66,67,68,69]. Digital biomarkers, such as eye-tracking (AUC 0.88) and breath sensors (90% accuracy), show potential for detecting Alzheimer’s and Multiple Sclerosis. Multi-modal systems (including smart home sensors, video, audio, and accelerometers) combine motor and cognitive monitoring. Although it achieves up to 80% accuracy, clinical integration is hindered by challenges in data interpretation and workflow adaptation. Future advances depend on data standards, interpretable AI, user-friendly clinical tools, and large-scale validation studies.

### 2.4. Digital Biomarkers

Movement Analysis: Wearables have reached 95% accuracy in detecting early Parkinson’s disease, with gait parameters indicating disease progression [70,71]. Voice and Speech Patterns: Early signs include changes in pitch, rhythm, and lower volume. Monitoring is non-invasive, but needs more validation. Cognitive Function Markers: Smartphone tests, eye-tracking, and smart home sensors show potential but face challenges in standardization [72].

### 2.5. Artificial Intelligence and Machine Learning in NND Monitoring

Real-time data collection: Wearables monitor heart rate, movement, sleep, and cognitive tasks [73,74]. Pattern recognition and predictive analytics: AI detects subtle shifts in gait, behavior, or physiology [75]. Personalized treatment plans: Optimized therapies tailored to patient profiles [76]. Early detection: AI identifies pre-clinical decline, enabling proactive intervention [77,78].

### 2.6. Remote Patient Monitoring and Telemedicine

AI-enabled platforms integrate wearable data into telehealth systems, supporting remote, continuous care for patients with mobility or geographic barriers. Proactive alerts enable timely interventions, reducing hospital visits and enhancing outcomes [78].

### 2.7. Vision-Based Approaches as Sensor-Based Technology

Vision-based approaches are increasingly recognized as part of sensor-based technologies, but they differ from traditional wearable and physiological sensors in their mode of operation. In sensor taxonomy, they are classified as non-contact digital sensors because they utilize optical devices, such as RGB, depth, or infrared cameras, to capture human motion, gait, or facial cues, unlike contact-based wearables like IMUs, EEG, or EMG electrodes [79,80,81,82,83,84,85,86,87,88,89]. These systems can produce overlapping data with wearable sensors—such as motion tracking comparable to accelerometers—while also providing richer spatial–temporal insights, which can make some wearables potentially redundant [90]. However, vision-based methods face specific challenges. Their outputs require complex computational models, intensive learning, and computer vision techniques, making interpretation less straightforward than with traditional wearable sensors [90,91]. Additionally, privacy concerns are a major limitation, as video-based monitoring is more sensitive than most wearable technologies and raises ethical issues related to consent and data protection [89]. Environmental factors, including lighting variability, occlusion, and camera placement, further limit reliability in real-world settings [91]. Despite these challenges, vision-based systems can complement wearables in multi-modal monitoring, boosting accuracy and scalability when integrated carefully into sensor ecosystems [91].

### 2.8. Clinical Implementation Progress

Validation Outcomes. Several studies have reported high accuracy rates in distinguishing patients with neurodegenerative diseases from healthy controls. An accuracy of 92.3%, a sensitivity of 90.0%, and a specificity of 100% were achieved for Parkinson’s disease screening using wearable sensors, and another such system was found to be 95% accurate in differentiating early, untreated Parkinson’s disease patients from healthy subjects [70]. Breath analysis has been used to differentiate between Multiple Sclerosis patients and controls with up to 90% accuracy [81]. Additionally, accuracy rates of 85% for Alzheimer’s disease compared to healthy subjects, 78% for Parkinson’s disease versus healthy subjects, and 84% for Alzheimer’s disease versus Parkinson’s disease have been achieved through breath analysis [82]. However, limitations exist, as many studies had small sample sizes or lacked rigorous validation methods, which restricts the generalizability of these results. Furthermore, performance metrics and methodologies varied widely across studies, making direct comparisons challenging.

### 2.9. Real-World Applications

Several studies have explored the potential of home-based monitoring technologies for real-world applications, especially in home-based or continuous monitoring settings. Smartphone platforms have demonstrated the feasibility of using a smartphone-based platform for remote monitoring of Parkinson’s disease and dementia symptoms in daily life [83]. Additionally, smart home sensors have shown their capability for long-term, unobtrusive monitoring of cognitive decline in real-world situations [84]. Wearable sensors can accurately detect symptoms of Parkinson’s disease using inertial sensors in free-living environments [84].

Potential benefits include real-world applications, which offer the chance for more ecologically valid assessments, as well as continuous monitoring, which could give clinicians more comprehensive and timely information about disease progression. However, challenges can include user acceptance, data privacy, and integration with existing clinical workflows, which remain significant hurdles.

### 2.10. Integration Challenges

Many challenges involve data interpretation, especially when processing and understanding substantial amounts of information from continuous monitoring technologies, which poses significant obstacles. The primary challenge in data interpretation from continuous monitoring is not just managing the volume of data but transforming raw streams into reliable, understandable, and context-aware insights that clinicians and patients can trust and utilize. Another challenge concerns advanced analysis techniques, highlighting the need for sophisticated methods like deep learning to effectively process and interpret sensor data for evaluating Parkinson’s disease and other NDDs [85]. Additionally, there is a lack of standardization. The wide variety of technologies, measurement protocols, and performance metrics used across studies complicates the comparison of results and the establishment of clinical guidelines. Furthermore, user acceptance remains an issue. Long-term adherence to monitoring protocols is critical, especially for wearable devices and home-based systems. Clinical workflow integration: Ensuring that the data produced by these technologies is actionable and meaningful in clinical settings is essential for their successful adoption [80].

Table 2 displays the various levels of technological readiness of sensor-based technologies for detecting neurodegenerative diseases. Wearable inertial sensors are the most validated and dependable tools for identifying early-stage Parkinson’s disease. A comprehensive review of 296 studies on wearable sensors for home monitoring of individuals with Parkinson’s disease, published in *NPJ Parkinson’s Disease*, highlighted their diagnostic sensitivity and clinical utility in various settings [92].

In contrast, other modalities, such as breath analysis, eye-tracking, and multi-modal systems, show strong potential for early detection but require more comprehensive clinical validation. Breath analysis, for example, has shown promise in detecting neurodegenerative diseases by analyzing exhaled volatile organic compounds. Similarly, eye-tracking technology is considered a promising tool for diagnosing and monitoring the progression of Parkinson’s disease. Multi-modal systems, which combine data from various sensors, have been studied for early detection of neurodegenerative diseases by utilizing brain MRI and wearable sensor data [93].

However, although these technologies show promise, they often lack thorough clinical validation and are mainly supported by proof-of-concept studies. For example, a study on an iPad-based eye movement assessment system for early Parkinson’s disease detection demonstrated its feasibility. However, it also highlighted the need for further validation with clinical-grade equipment [94].

These observations emphasize the need to carefully interpret reported accuracy metrics. Factors such as study size, patient demographics, validation methods, and the environment in which the technology is used greatly influence the generalizability and clinical usefulness of the results. Without this contextual information, it is hard to assess whether these technologies are ready for widespread clinical adoption.

## 3. Challenges and Future Directions

Despite the summarized challenges and limitations listed in Table 3, artificial intelligence (AI) is poised to revolutionize our understanding, detection, and treatment of neurodegenerative diseases. It promises significant advances in medical diagnosis, disease detection, progression modeling, and personalized treatment strategies. Real-world case studies demonstrate AI’s transformative potential in clinical settings; however, implementing it poses challenges, including data privacy issues, the need for model interpretability, potential biases, and technological limitations.

Overcoming these obstacles requires a collaborative effort from AI researchers, healthcare providers, and policymakers to ensure that AI is used ethically and securely in this field. Despite these challenges, integrating AI and machine learning into the monitoring of neurodegenerative diseases offers great promise. Ongoing advancements in wearable technology, data analytics, and interdisciplinary collaboration will drive innovation, enhance patient management, and ultimately improve the quality of life for individuals with these debilitating conditions. The primary goal is to seamlessly integrate technology and clinical care, empowering patients, and healthcare providers in the fight against neurodegenerative diseases, leading to improved diagnostic accuracy and a deeper understanding of disease progression (See Table 3, which summarizes challenges, limitations, future directions, and research opportunities).

## 4. Conclusions

The review highlights how technological advances are transforming the diagnosis and monitoring of neurodegenerative diseases, including Alzheimer’s, Parkinson’s, Huntington’s disease, and amyotrophic lateral sclerosis. Over the past decade, advances in sensor technology, nanomaterial-based biosensors, and artificial intelligence have shifted the focus from traditional, symptom-based assessments to methods that are objective, continuous, and biomarker-based. This shift is crucial because it enhances diagnostic accuracy and enables the earlier detection of disease processes, often before clinical symptoms manifest.

A wide variety of new sensors are now used in both research and clinical settings. Wearable and implantable devices can track physiological and behavioral signs, such as gait, tremor, speech, sleep, and autonomic function, in real-world surroundings, capturing changes often missed during brief clinical visits. Meanwhile, non-invasive biosensing platforms—such as graphene-based field-effect transistors, electrochemical sensors, and optical devices—have demonstrated the ability to detect pathological proteins, including amyloid-β, tau, and α-synuclein, in peripheral fluids with impressive sensitivity. The use of nanomaterials in these devices has enhanced their selectivity and stability, paving the way for point-of-care testing that may someday replace or complement invasive diagnostic methods, such as lumbar punctures.

Artificial intelligence plays a crucial role in interpreting the vast, multi-modal datasets generated by these tools. Machine learning algorithms can identify subtle patterns in high-dimensional data, supporting tasks such as classifying disease subtypes, predicting disease progression, and customizing treatments for individual patients. In clinical trials, AI-driven analysis of sensor-based data enhances patient stratification, facilitates more efficient trial design, and may increase the likelihood of successful therapeutic outcomes.

The implications for both clinical care and research are significant. Clinically, these technologies support precision medicine by allowing earlier interventions, real-time monitoring of treatment effects, and more tailored management strategies. For researchers, sensor-based systems offer unique opportunities to collect large-scale, long-term datasets that reveal the interaction between biological, behavioral, and environmental factors in neurodegeneration. These data not only enhance our understanding of disease mechanisms but also lay the groundwork for developing integrated models that link basic neuroscience to clinical outcomes.

Despite their promises, integrating sensor technologies into standard care faces challenges. Rigorous validation is essential to guarantee accuracy and consistency across diverse populations. Data formats and protocols need standardization to ensure compatibility across platforms and with existing clinical systems. Ethical issues, such as patient privacy, informed consent, data ownership, and cybersecurity, also require attention, especially as continuous monitoring generates extremely sensitive health data. Costs and accessibility pose additional obstacles, raising key questions about how to ensure equitable adoption across various healthcare settings.

Looking toward the future, the success of these innovations will rely on close collaboration among neuroscientists, engineers, clinicians, computer scientists, ethicists, and policymakers. Regulatory frameworks will also need to adapt to evaluate and approve new devices and algorithms at the same pace as technological progress. As digital health ecosystems grow, integrating sensor-based platforms with telemedicine, cloud computing, and personalized treatment strategies could enable remote monitoring and interventions, decreasing healthcare burdens and enhancing patients’ quality of life. Furthermore, as artificial intelligence becomes more transparent and interpretable, clinicians will be able to convert complex sensor data into actionable insights for personalized decision-making.

Ultimately, sensor-based technologies should no longer be viewed as optional extras, but as essential components of a next-generation framework for diagnosing, monitoring, and understanding neurodegenerative diseases. By providing continuous, objective, and biomarker-informed assessment, these tools have the potential to transform both research and clinical practice. Overcoming current challenges in validation, standardization, ethics, and equitable access will be essential. Still, the pace of innovation strongly indicates that precision neurology—characterized by early detection, personalized treatments, and better outcomes—is becoming a feasible goal for patients with these devastating disorders.

## Figures and Tables

**Table 2 sensors-25-05476-t002:** Technology platforms, detection accuracy, early detection capabilities, and clinical validation status.

Technology Platform	Detection Accuracy	Early Detection Capability	Clinical Validation Status
Wearable inertial sensors	Approximately 95% accuracy for early Parkinson’s disease detection; over 90% accuracy for Parkinson’s disease symptom detection in free-living environments.	Demonstrated capability for detecting early-stage Parkinson’s disease.	Validated in multiple studies, including real-world settings.
Smartphone-based sensors	Approximately 98% accuracy in step length estimation, 94% accuracy in identifying gait changes for Parkinson’s disease.	Showed potential for early Parkinson’s disease detection through gait analysis.	Limited clinical validation, mostly proof-of-concept studies.
Breath analysis	Up to 90% accuracy for Multiple Sclerosis detection; 85% accuracy for Alzheimer’s disease vs. healthy, up to 78% for Parkinson’s disease vs. healthy.	Demonstrated potential for early-stage detection.	Limited clinical validation, mostly experimental studies.
Blood-based biomarkers (ultrasensitive detection)	High sensitivity and specificity were reported, but specific metrics were not provided.	Showed potential for early Alzheimer’s disease detection.	Promising results, but limited large-scale clinical validation.
Eye-tracking	Receiver operating characteristic area under the curve of 0.88 for differentiating Parkinson’s disease patients from controls.	Demonstrated potential for early cognitive decline detection.	Limited clinical validation, mostly experimental studies.
Smart home sensors	Although specific accuracy metrics were not reported, the system showed potential for monitoring long-term mild cognitive impairment.	Demonstrated capability for detecting early signs of cognitive decline.	Limited clinical validation, mostly proof-of-concept studies.
Multi-modal systems	Up to 80% accuracy reported for combined sensor approaches.	Showed potential for comprehensive early detection.	Limited clinical validation, mostly experimental studies.

**Table 3 sensors-25-05476-t003:** Summary of challenges, limitations, future directions, and research opportunities.

Disease and Setting	Challenges	Limitations	Future Directions	Research Opportunities
Alzheimer’s (Clinic)	Subtle cognitive/motor decline is challenging to capture due to variability in testing environments.	Cognitive tests are often episodic, rather than continuous; imaging is also costly.	Multi-modal in-clinic sensors (eye-tracking, digital pen, EEG).	Early detection of mild cognitive impairment via digital biomarkers.
Alzheimer’s (Home)	Adherence to wearables and noise from daily routines.	Smart-home monitoring is costly and raises privacy concerns.	Passive monitoring (speech, mobility, sleep) using IoT.	Digital phenotyping of early memory and language decline.
Parkinson’s (Clinic)	Tremor/rigidity fluctuates; stress and meds alter readings.	Single-time-point measurements miss variability.	Digital gait labs and wearable sensors are available in the clinic.	Sensor-validated motor scoring aligned with MDS-UPDRS.
Parkinson’s (Home)	Continuous tremor and gait monitoring → large, noisy datasets.	Device heterogeneity, patient compliance.	Smartphone-based tapping/voice apps; continuous gait sensors.	Prodromal PD detection: real-world treatment–response biomarkers.
ALS (Clinic)	Rapid progression, heterogeneity of symptoms.	Clinical measures (ALSFRS-R) are subjective and infrequent.	Sensor-based speech and respiratory testing.	Digital endpoints for respiratory decline detection.
ALS (Home)	Difficulty in sustained sensor use as the function declines.	Limited accessibility/adapted devices.	Voice analysis apps and respiratory wearables for home use.	Longitudinal tracking of speech/motor decline for trials.
Huntington’s (Clinic)	Movements are variable; psychiatric symptoms are less quantifiable.	Imaging/clinical tests are limited in frequency.	Motion-capture systems; digital cognitive tests.	Quantifying subtle motor/cognitive onset before diagnosis.

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
