# Peer review of "Innovative Sensor-Based Approaches for Assessing Neurodegenerative Diseases: A Brief State-of-the-Art Review"

_sensors, 2025, doi:10.3390/s25175476_

Round 1
Reviewer 1 Report
Comments and Suggestions for Authors
In general, I found the paper quite rfepoetitive and a bit disorganized. Several sections and discussions were repetitive.
As a whole, the article does not seem to add to the general knowledge in the field. This article (or a shorter form) looks more suitable for an article in the science section of a newspaper than a scientific journal.
A few detailed comments are attached.

Author Response
Thank you for taking the time to review our manuscript. Please see the highlighted areas in the manuscript. We updated a couple of areas and eliminated redundant sections. Thank you for pointing this out. We agree with the comment. Therefore, we have made significant changes starting on page 89 of the manuscript. In section 2, we changed the title to "Advances in Technology for NND Monitoring." We made edits to the conclusion and removed redundant areas. We also streamlined the information, making it shorter and easier to understand, as seen on page 95. Additionally, we added a section discussing vision-based approaches that was not previously addressed, on page 175. We provided more details about Table 2 regarding technology readiness. We also revised the conclusion and eliminated redundancies. All the revisions are highlighted in yellow ink.
Reviewer 2 Report
Comments and Suggestions for Authors
This paper presents a review of sensor-based assessment of neurodegenerative diseases. The topic itself is timely and interesting, and it gives an okay narrative and reading experience as an entry-level domain overview.
But still, some big concerns remain:
1. The review largely lists devices, accuracies, and reported statistics from individual studies, but there is minimal critical comparison, discussion of methodological quality, or synthesis of trends and limitations. There is little exploration of why certain sensor modalities work better, under what conditions they fail, and how findings translate to clinical adoption.
2. The paper talks extensively about AI-related aspects, but does not mention any advances in video-based assessment technologies, especially in Parkinson’s disease. It could be strengthened by discussing some emerging video-based AI methods for Parkinson’s assessment, for example
Zhang, H., Ho, E. S., Zhang, F. X., & Shum, H. P. (2022). Pose-based tremor classification for Parkinson’s disease diagnosis from video. MICCAI, pp. 489–499.
Liu, W., Lin, X., Chen, X., Wang, Q., Wang, X., Yang, B., ... & Lin, Y. (2023). Vision-based estimation of MDS-UPDRS scores for quantifying Parkinson's disease tremor severity. Medical Image Analysis, 85, 102754.
The authors should clarify whether such vision-based approaches are considered “sensor-based” in their definition, and if so, how they compare with wearable sensors. If not, could they pose a practical challenge to current sensor-based methods given their easier deployment?
Also some minor points:
1. The focus in Section 2 is a little bit lost and contains long, generic overviews of all emerging technologies, which could be condensed or moved to a short background paragraph.
2. Evidence quality and context gaps. Performance metrics such as “up to 95% accuracy” are often quoted without reporting dataset sizes, patient demographics, validation methods, or whether the results came from controlled lab settings or real-world deployments. Without this context, readers cannot judge generalizability or readiness for clinical translation.
3. Several sections repeat similar statements about the benefits of wearable sensors and AI integration, which could be streamlined to avoid redundancy and improve readability.
4. The challenges section could be expanded to discuss technical barriers such as calibration, multimodal data fusion, and cost-effectiveness, rather than focusing mainly on generic issues like privacy and interoperability.
5. Some abbreviations are inconsistent, such as “LoT” instead of “IoT” and “neurogenerative” instead of “neurodegenerative” in the abbreviations table.
Overall, I’d recommend a major revision before the paper is ready to be presented to the audience.
Author Response
|
We have accordingly revised or modified the document to emphasize the feedback below. Thank you for providing the information on vision-based approaches. We added a section to the paper that addresses these areas, along with additional citations that are critical to the field. We discussed how these are compared to wearable sensors. We appreciated this perspective. On the evidence of quality and context gaps, we understand. We included further details with the following statement: "These observations emphasize the importance of carefully interpreting reported accuracy metrics. Factors such as study size, patient demographics, validation methods, and the environment in which the technology is significantly used influence the generalizability and clinical usefulness of the results. Without this contextual information, it becomes challenging to determine how ready these technologies are for widespread clinical adoption." We provided additional information to address the areas of concern. We even included a table that summarizes the challenges, limitations, future directions, and research opportunities.
Thank you for pointing out the redundancy issue. We agree with this comment. Therefore, we have made significant changes starting on page 89 of the manuscript. In Section 2, we changed the title to "Advances in Technology for NND Monitoring."
We made changes to the conclusion and eliminated areas of redundancy. We also streamlined the information provided, making it shorter and easier to understand. This can be seen on page 95. We added a section that discussed vision-based approaches not previously addressed on page 175. We provided more information about Table 2 on technology readiness. We also revised the conclusion and eliminated areas of redundancy. All the revisions were highlighted in yellow ink.
|
Reviewer 3 Report
Comments and Suggestions for Authors
In this review, the authors introduced the innovative sensor technologies, wearable and sensor-based monitoring technologies, and AI and machine learning for assessing neurodegenerative diseases. It provided a wealth of helpful information about this subject. However, the content of this paper needs to be enhanced to be recognized as a reviewing paper.
- Lines 89 and 430, no citation or description in the text.
- The section of “2. Categories of Innovative Technologies” only involved some common information. The section could be deleted.
- There are two key points about this paper.
Lines 289-292, “ However, challenges remain with using these multi-modal monitoring systems, including data integration, interpretation, and practical implementation in clinical settings, which present significant hurdles for these systems.”
Lines 408-411, “ Many challenges involve data interpretation, particularly the processing and understanding of extensive information from continuous monitoring technologies, which present significant obstacles.”
Please provide more literature to illustrate the statement of these paragraphs.
- In the Abstract, the authors mentioned “We discussed challenges, limitations, future directions, and research opportunities.” Please provide a table to illustrate these statements and cite all related literature.
- The style of references did not correspond to the requirements of this journal.
This review contains many valid points, but its presentation is not well organized and, therefore, difficult to read. Please revise the content in detail.
Author Response
Thank you for reviewing our manuscript. We have revised or modified the document to emphasize the points mentioned below. We updated the information on lines 89 and 430, with all additions and changes highlighted in yellow ink. We addressed lines 289-292, 408, and 411, respectively. Due to the extensive revision, some information may have shifted to different sections. We included additional details to address the areas of concern and added a table summarizing the challenges, limitations, future directions, and research opportunities. We also revised the section 2 title to 'Advances in Technology for NND Monitoring' and streamlined the content for easier reading.
We made adjustments to the conclusion by removing redundancy and simplifying the information to make it shorter and easier to understand, as shown on page 95. Additionally, we added a section discussing vision-based approaches that were not previously covered on page 175. We provided more details about Table 2 on technology readiness, and the revised sections are highlighted in yellow. All changes are clearly marked.
|
We provided a total of nine brand new references that discussed all the revisions to our manuscript. In addition, we updated the references to align with the journal’s requirements. We greatly appreciated the reviewers’ feedback on ways to improve our manuscript. |
Round 2
Reviewer 1 Report
Comments and Suggestions for Authors
The article has improved significantly in its structure.
Author Response
We value your feedback. It played a key role in revising the overall structure of our manuscript.
Reviewer 2 Report
Comments and Suggestions for Authors
The authors have satisfactorily addressed my concerns. I would only suggest one minor point: the new table appears to be inserted as an image. It would be preferable to keep it in an editable table format to ensure readability, especially when zooming or enlarging.
Author Response
Thank you very much. We have updated Table 3 to a more readable and adjustable format. We appreciate you taking the time to review our manuscript and share your feedback. Your time is greatly appreciated.
Reviewer 3 Report
Comments and Suggestions for Authors
All problems have been addressed adequately. However, the style of references needs to be rechecked..
Author Response
Thank you very much for your time and feedback. We have reviewed the references to ensure they meet the journal’s specifications. We appreciate your effort and the time you dedicated to reviewing our manuscript.